# Based on FCN and DenseNet Framework for the Research of Rice Pest Identification Methods

He Gong [1,2,3,4], Tonghe Liu [1], Tianye Luo [1], Jie Guo [1], Ruilong Feng [1], Ji Li [1], Xiaodan Ma [1], Ye Mu [1,2,3,4], Tianli Hu [1,2,3,4], Yu Sun [1,2,3,4], Shijun Li [5,6], Qinglan Wang [7] and Ying Guo [1,2,3,4,*]

1   College of Information Technology, Jilin Agricultural University, Changchun 130118, China
2   Jilin Province Agricultural Internet of Things Technology Collaborative Innovation Center, Changchun 130118, China
3   Jilin Province Intelligent Environmental Engineering Research Center, Changchun 130118, China
4   Jilin Province Colleges and Universities and the 13th Five-Year Engineering Research Center, Changchun 130118, China
5   College of Information Technology, Wuzhou University, Wuzhou 543003, China
6   Guangxi Key Laboratory of Machine Vision and Inteligent Control, Wuzhou 543003, China
7   Jilin Academy of Agricultural Sciences, Changchun 130033, China
*   Correspondence: guoying@jlau.edu.cn

**Abstract:** One of the most important food crops is rice. For this reason, the accurate identification of rice pests is a critical foundation for rice pest control. In this study, we propose an algorithm for automatic rice pest identification and classification based on fully convolutional networks (FCNs) and select 10 rice pests for experiments. First, we introduce a new encoder–decoder in the FCN and a series of sub-networks connected by jump paths that combine long jumps and shortcut connections for accurate and fine-grained insect boundary detection. Secondly, the network also integrates a conditional random field (CRF) module for insect contour refinement and boundary localization, and finally, a novel DenseNet framework that introduces an attention mechanism (ECA) is proposed to focus on extracting insect edge features for effective rice pest classification. The proposed model was tested on the data set collected in this paper, and the final recognition accuracy was 98.28%. Compared with the other four models in the paper, the proposed model in this paper is more accurate, faster, and has good robustness; meanwhile, it can be demonstrated from our results that effective segmentation of insect images before classification can improve the detection performance of deep-learning-based classification systems.

**Keywords:** pest identification; FCN; DenseNet; attention mechanism

## 1. Introduction

Climate, ecology, natural catastrophes, and many other factors have profound impacts on the production of grains, with insect damage being one of the major factors affecting crop productivity. Numerous crops, including wheat, maize, and rice, have lower yields as a result of agricultural pests. Hence, to effectively control pests, it is necessary to predict the occurrence trend, quantity, population dynamics, and potential damage of pests, while real-time and accurate forecasting is very important. The correct identification and classification of rice pests is a prerequisite for the prevention and management of rice pests. Insect experts or insect taxonomists typically carry out traditional insect classification and identification work based on specialized expertise, research experience, or reference maps. However, a lot of time and effort are required by this method, which has generally low work efficiency and extremely unstable accuracy [1]. The development of an automatic identification and classification system of pests will remarkably reduce the labor intensity of plant protection personnel and improve the accuracy of forecasting, thereby reducing the loss of rice yield.

With the continuous development of Internet technology, the use of computer vision technology for pest identification has gradually been widely studied. Many works in the

literature have applied the traditional machine vision technique to pest identification [2,3]. However, this method not only has difficulty in meeting actual needs, but also the generalization ability and robustness of the model are relatively poor. In the past few years, with the continuous development and updating of artificial intelligence technologies, such as deep learning technology and big data technology, more possibilities have emerged for examining the pest identification problem [4]. Rice is considered one of the most important food crops, and its yield directly affects many issues, such as food security. Rice will inevitably be affected by different pests during the growth process. Thus, controlling the scale of pests and diseases at an appropriate time can reduce the amount of pesticide spraying and avoid a large reduction in rice production. Therefore, the accurate identification of pests has become an important basis for pest control, which renders the investigation of the rice pest identification technology particularly important.

Currently, the scientific community has more options to create increasingly automated systems that can accurately recognize objects of any kind thanks to the astonishing progress of picture classification [5]. Yang et al. [6] used the shape of the insect image, extracted the features with color, and then utilized the radial basis neural network to classify it with an accuracy rate of 96%. Zhang et al. [7] proposed a faster R-CNN framework that was composed of automatic recognition algorithms for two types of insects. The recognition accuracy rate of this model reached 90.7%. Chen et al. [8] reported the combination of machine learning and convolutional neural networks to identify five kinds of corn pests in the northeast cold region. Cheng et al. [9] introduced a deep convolutional neural network. After the feature extraction was performed with color, the accuracy rate for classification using a radial basis neural network was 97.6%. Based on the standard convolutional neural network, Sun et al. [10] introduced the attention mechanism and created a convolutional neural network model based on the attention mechanism to recognize soybean aphids. However, the above methods still have defects, such as insufficient sample size of data, complex data preprocessing, insufficient feature extraction, large fluctuation of model fitting degree, similar target features, etc. Moreover, the features vary greatly among different insects and there are many types of rice pests, the above methods do not have relevant parameter adjustment and optimization for rice pests, and the models do not have good generalization, so they are not applicable to the classification of rice pests.

Segmentation is the basic stage of recognition and classification. The main purpose is to filter the image noise that is generated when the image is collected, remove the redundant background information of the target image, and extract the target object in a targeted manner to extract accurate, concise, and expressive feature information in the follow-up work. Before the advent of fully convolutional networks (FCN) [11], there were also some traditional methods for semantic segmentation, such as normalized cut [12], structured random forests [13], and SVM [14]. FCN is the first network that delves semantic segmentation to the pixel level, because FCN avoids the problem of repeated storage and computational convolution due to the use of pixel blocks; therefore, FCN is more efficient than traditional networks based on conventional convolutional neural networks (CNNs) [15] for segmentation, and there is no limit on the input image size. DenseNet proposes a more radical dense connection mechanism: interconnecting all layers. Specifically, each layer accepts all the previous layers as its additional input. Due to the dense connection approach, DenseNet improves the back propagation of gradients, making the network easier to train and with smaller and more computationally efficient parameters. To better solve the pest classification problem, we proposed a simple and efficient full convolutional network based on FCN and introduced an encoder–decoder CRF network, long-hop and short-hop connection mechanisms to solve the problems of the poorly segmented and detail insensitive FCN, and an efficient channel attention (ECA) mechanism [16] of the DenseNet network to further enhance the performance of the network. While strengthening the extraction of insect features, the extraction of invalid background features is inhibited, so as to improve the identification accuracy and generalization ability of the network and ensure the efficient and accurate identification of insect pests.

## 2. Materials and Methods

### 2.1. Experimental Data Set

The source of the data stems from the intelligent insect forecasting lamp, and the location is multiple rice fields in Shuangyang District and Jiutai District, Changchun City, Jilin Province, from July to September 2021. The intelligent insect forecasting lamp was automatically turned on after sunset every day, and the trapped insects were heated and inactivated by the electric heating plate to make it easier to shoot. Finally, the white conveyor belt was used to tile the insect corpses. The rolling time can be set according to the period of a high incidence of the insect pests and the conveyor belt can be manually and remotely controlled to ensure that the data are not repeated and are not unreliable. The industrial camera was used for vertical shooting, and the shooting was synchronized with the rolling of the transmission belt. The model of the industrial camera was MV-CE120-10UM/UC 1/1.7'CMOS MV-CE120-10UC with 12 million pixels, while the manufacturer was HIKROBOT. It can also be shot manually and remotely at any time. The resolution was $4024 \times 3036$, the unit was px, and 4236 original pictures were obtained. The captured insect images were automatically uploaded to the cloud database for subsequent processing. Part of the original images are depicted in Figure 1.

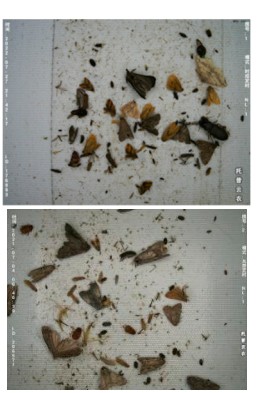 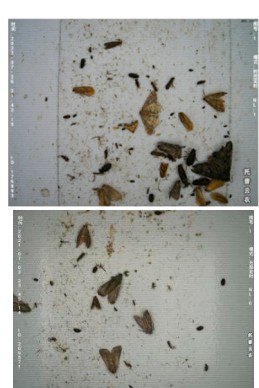 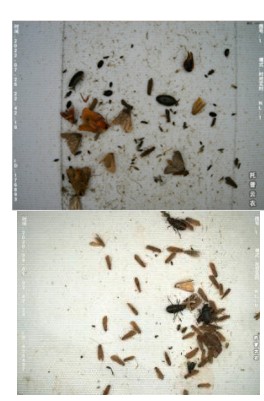 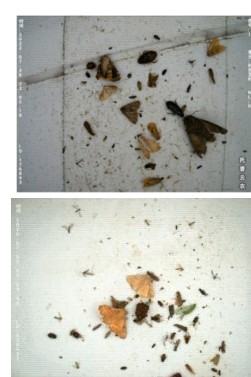

**Figure 1.** Part of the original images.

Due to the impact of the wild biodiversity, collection locations, and other factors, most of the insect populations are relatively small. In order to ensure the reliability of the experiment and the accuracy of the classification, according to the data collection of the actual local insect species, the classification was carried out with reference to the 2022 version of the "Catalogue of Biological Species in China". The relevant experts from the Animal Science and Technology College of Jilin Agricultural University also provided valuable help. The most harmful insect pests are *Chilo suppressalis* (*Lepidoptera*:Pyralidae), *Naranga aenescens* (*Lepidoptera*:Noctuidae), *Cnaphalocrocis medinalis* (*Lepidoptera*:Pyralidae), *Nilaparvata lugens* (*Homoptera*:Delphacidae), *Agrotis ypsilon* (*Lepidoptera*:Noctuidae), *Gryllotalpa* sp. (*Orthoptera*:Grylloidea), *Mythimna separata* (*Lepidoptera*:Noctuidae), *Helicoverpa armigera* (*Lepidoptera*:Noctuidae), *Gryllidae* (*Orthoptera*:Gryllidae), and *Holotrichia diomphalia* (*Coleoptera*:Melolonthidae) as the target insects. Since the traps are based on the phototaxis of insects, the identification studies in this work are for adults. The insects in the original image have great uncertainty, while the repeated and wrong images were removed through manual screening. In order to facilitate the display of the pest images, the target insects in the acquired data were marked and classified. The tagging work was completed by using the software Visual Object TaggingTool(VoTT)v2.2.0 developed by Microsoft Corporation, and the storage format was PascalVOC. The experimental data set had a total of 2225 pictures. Among them, there are 422 pictures of *C. suppressalis*, 356 pictures of *N. aenescens*, 335 pictures of *C. medinalis*, 256 pictures of *N. lugens*, 239 pictures of *A. ypsilon*, 123 pictures of *G.* sp, 366 pictures of *M. separata*, 189 pictures of *H. armigera*, 125 pictures of

Gryllidae, and 128 pictures of *H. diomphalia.* Partial images of some rice pests selected from the test set are shown in Figure 2.

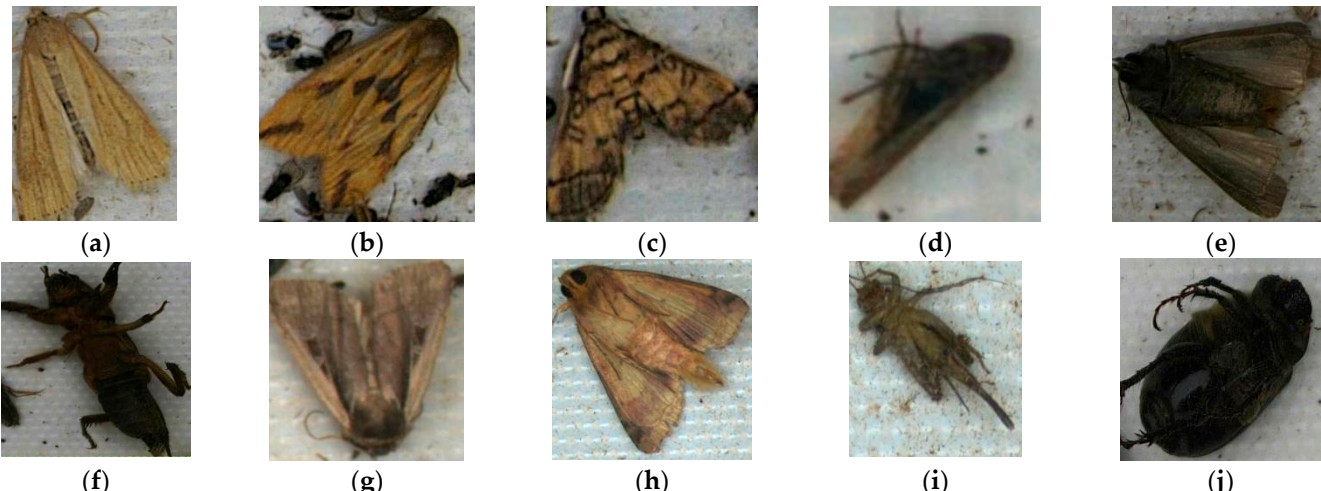

(**a**)  (**b**)  (**c**)  (**d**)  (**e**)

(**f**)  (**g**)  (**h**)  (**i**)  (**j**)

**Figure 2.** Examples of pest images: (**a**) *C. suppressalis*, (**b**) *N. aenescens*, (**c**) *C. medinalis*, (**d**) *N. lugens*, (**e**) *A. ypsilon*, (**f**) *G.* sp, (**g**) *M. separata*, (**h**) *H. armigera*, (**i**) Gryllidae, (**j**) *H. diomphalia*.

### 2.2. Data Augmentation

In deep learning, for the training of convolutional neural networks, a large amount of data sets is often required. Otherwise, various phenomena such as overfitting and low recognition accuracy will take place. However, under the existing conditions, due to the difficulty in collecting rice pest data sets, and the lack of existing rice pest data, some data expansion methods were used here to achieve the purpose of increasing the data set. Particularly, shift, scale, rotation, flip, noise, brightness, and other data expansion methods were utilized to expand the original data set by 10 times, which corresponds to 4220 pictures of *C. suppressalis*, 3560 pictures of *N. aenescens*, 3350 pictures of *C. medinalis*, 2560 pictures of *N. lugens*, 2390 pictures of *A. ypsilon*, 1230 pictures of *G.* sp, 3660 pictures of *M. separata*, 1890 pictures of *H. armigera*, 1250 pictures of Gryllidae, and 1280 pictures of *H. diomphalia*. According to the ratio of 6:2:2, these were divided into training data, verification data, and test data.

### 3. Model Refinement

By using the FCN algorithm as the basic framework, the DenseNet as the feature extraction network of the FCN algorithm, and introducing the channel attention mechanism, a rice pest recognition algorithm was proposed. We named the proposed model in this paper FCN-ECAD for ease of presentation and use. In Figure 3, the methodological framework of this study is described and illustrated and in the follow-up is thoroughly discussed.

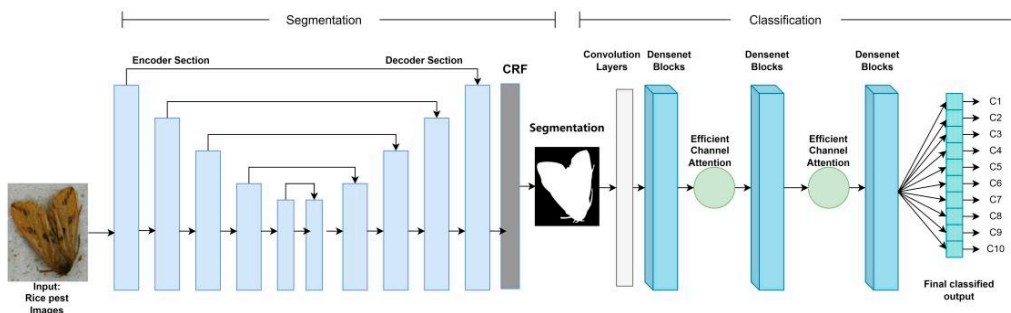

**Figure 3.** Schematic diagram of the proposed deep learning framework for image segmentation and classification of rice pests.

### 3.1. Feature Extraction Based on Encoder–Decoder Network

Figure 4 depicts the network's encoder and decoder parts, each of which has five different successive phases. Each stage consists of a cascaded layer, a ReLU activation layer, a convolutional layer with a 3 × 3 kernel size, and a series of skip routes. In the final step, there are 1024 convolutional filters instead of the initial 64. Moreover, instead of using the typical short jump connections, a sequence of jump pathways with both long and short jump connections was created. Nonlinearities were also added by using the ReLU activation module, which might hasten the network's training.

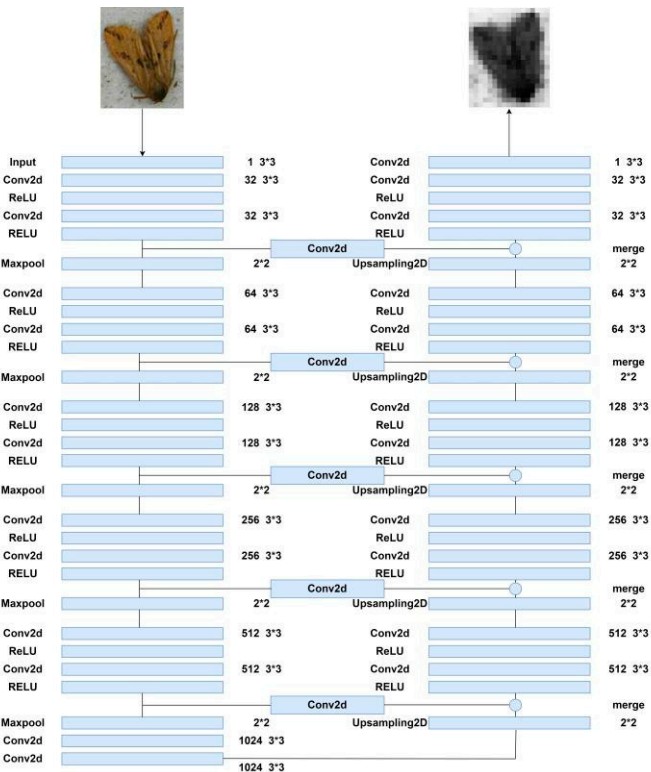

**Figure 4.** Deep convolutional encoder–decoder network structure diagram.

Additionally, the downsampling function was carried out by the encoder component using a max pooling module. As indicated in the equation, the decoder portion passes the pooled index to the appropriate upsampling layer after a feature vector has been retrieved from the input picture by employing a convolutional layer and downsampling by half using a max pooling module (1).

$$Y_i = U(F(I : r) : d) \tag{1}$$

where the final output, *F* denotes the downsampling feature map, *r* represents the RELU activation function, *d* stands for the downsampling module, and U is the upsampling module.

The decoder then applied an upsampling layer and multiplied the sample size by a factor of 2 to the feature vector from the preceding layer. In order to offer rich information, prevent gradients from dissipating, and restore the lost feature information, the matching output feature maps of the matched encoder component were next concatenated with these feature vectors. With a convolutional layer that has a 1 × 1 kernel and a softmax module, the decoder component was finished. The projected split was discovered to correspond to the class with the highest probability for each pixel by using a softmax classifier and the probability output from the N-channel image, as demonstrated by Equation (2).

$$P(\mathrm{y} = \mathrm{i}\,|\,x) = \frac{e^{x^T w_i}}{\sum\limits_{n=1} e^{x^T w_n}} \tag{2}$$

where $x$ is the feature map, $w$ refers to the kernel operator, and $n$ represents the number of classes.

### 3.2. Long-Hop and Short-Hop Connections

Figure 4 depicts the usage of the skip route of both long-hop and short-hop connections. As a reinforcement learning strategy for effective feature extraction, the system used short-hop connections to create very deep FCNs. Shortcut connections were also used to accelerate feature extraction and learning by using $2 \times 2$ convolutional layers. The method utilized a variety of skip pathways to hierarchically integrate downsampled and upsampled features and bring the encoder feature map's semantic level closer to that of the decoder. The spatial information that was lost during downsampling was replaced in the long-hop connections utilized for the upsampling step by the extracted features.

### 3.3. CRF

The CRF algorithm [17] is a standard algorithm that is widely used in edge detection. The CRF algorithm was introduced to ensure contour refinement and insect boundary localization to improve classification performance.

### 3.4. Feature Extraction Network DenseNet

In this work, DenseNet was adopted for insect feature extraction because the best performance based on the ImageNet classification task is provided. The TOP-1 of those popular pretrained models is summarized in Ref. [18] and the comparison results are displayed in Figure 5. As can be observed, DenseNet outperformed other pretrained models. Therefore, DenseNet was chosen as the feature extraction model in this work.

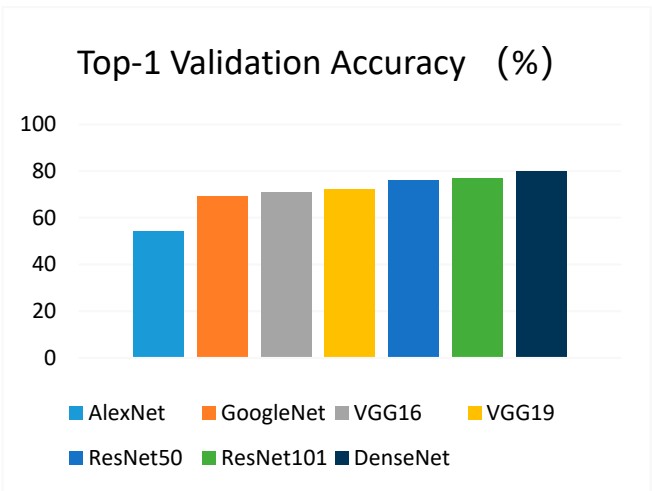

**Figure 5.** Comparison of the pretrained models.

A deeper network with dense connections was created via DenseNet. The dense block layer, which aimed to maximize information flow between network layers, is considered the most crucial component of the architecture [19]. Each layer in this architecture receives input from all layers before it and passes the feature mappings to all layers after it. The efficient transmission of the earlier features to the later ones for automatic feature reuse is made possible by these brief connections between the layers near the input and output. As a result, this network topology may be trained more precisely and effectively, and it can be utilized to extract more general and significant properties. Some works in the

literature have merged the features that were previously derived from each layer and then reprocessed the features [20–22]. This technique, which is just a basic concatenation of various feature maps, was not meant to promote feature reuse between layers. As a result, as illustrated in Figure 6, all prior levels were taken into consideration as input layers rather than integrating all feature maps.

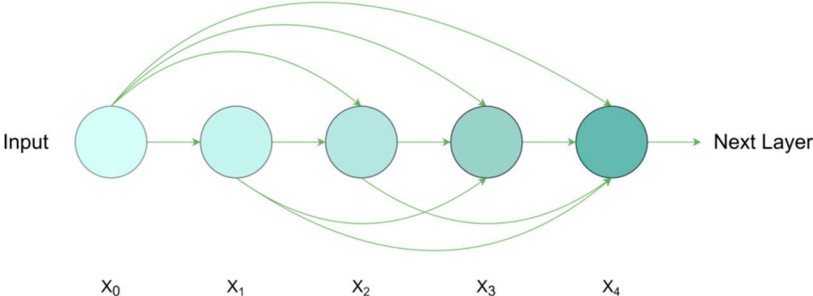

**Figure 6.** Operation of densely connected convolution, the dense block layer.

Therefore, unlike some traditional network structures, there are $l(l + 1)/2$ connections instead of only $l$ connections in the $l$ layer. In this way, the input feature map of layer l can be calculated based on the previous layers.

$$x_l = T_l([x_0, x_1, ..., x_{l-1}]) \tag{3}$$

where $[x_0, x_1, \dots x_{l-1}]$ represents the concatenation of layer 0, layer 1, and layer $l-1$ feature maps, respectively. In addition, the feature maps were concatenated dimensionally rather than using the pointwise sum as a reference. In the nonlinear transformation, (1) convolution, pooling [23], and rectified linear units [24], among other processes, can be found in $T_l(\cdot)$. For concatenated operations, each dense block consists of several sets of $1 \times 1$ and $3 \times 3$ convolutional layers with the same padding. Despite using tightly connected patterns, this structure employed fewer parameters than the conventional convolutional networks. In fact, this network architecture decreased the number of feature maps needed for network layers and did away with the need to learn redundant information. The parameter efficiency was thereby greatly increased. On the other hand, consecutive cascades of various layers necessitate access by each layer to the gradients from the initial input data, as well as the loss function. This quick access enhanced the communication across layers and lessened the disappearing gradient issue. This strategy of feature reuse aided in the development of a deeper network architecture and the extraction of deep semantic relations.

### 3.5. Channel Attention Mechanism

The attention mechanism was added to concentrate on extracting the target insect features in the image in the complex environment of the rice field, which was impacted by the wild biodiversity and diverse backgrounds. In order to boost the efficiency of the deep convolutional neural networks, the ECA-Net attention module was used here, which was a lightweight module. More specifically, it concentrated on the extraction of significant features and suppressed the unimportant features by combining the depth and spatial information of feature maps with an effective attention module. In challenging environments, the extraction can effectively increase the field insects' recognition accuracy. The ECA-Net diagram is shown in Figure 7. First, the input feature map underwent global average pooling, whereas each channel's feature layer was represented by a single value. To obtain the interdependence between each channel and their relationship, a one-dimensional convolution with a size of k was used in the second step. A Sigmoid activation function was then added for normalization, and finally, the weights of each channel were multiplied with the input feature map to strengthen the extraction of significant features.

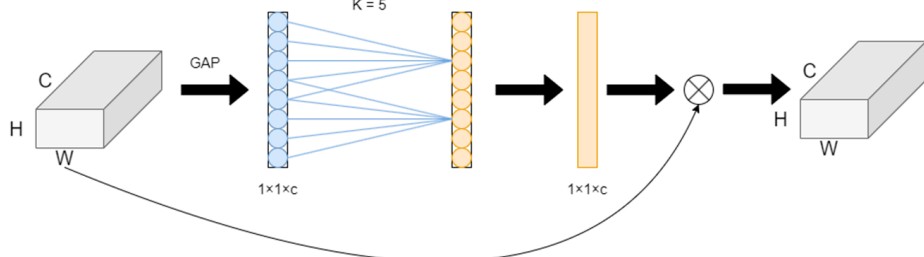

**Figure 7.** ECA-Net network architecture: the global pooling layer (GAP) obtains aggregated features and ECA generates channel weights by performing one-dimensional convolution of size *k*, where the value of *k* was determined by the channel dimension *C*.

ECA used a one-dimensional convolution of size *k* to interact across channels instead of a fully connected layer, which can effectively reduce the amount of computation and complexity of the fully connected layer, and then generate weights for each channel, namely

$$\omega = \delta\left(CID_k(y)\right) \tag{4}$$

In the formula, $\omega$ denotes the channel weight, $\delta$ is the Sigmoid activation function, and *CID* represents a one-dimensional convolution. The existence of more channels of the input feature map leads to a greater value of local interaction. As a result, the value of *k* is proportional to the number of channels, *C*. The dimension-related function adaptively determines the value of *k*, namely

$$C = 2^{(\gamma \cdot k - b)} \tag{5}$$

$$k = \left| \frac{\log_2(C)}{\gamma} + \frac{b}{\gamma} \right|_{odd} \tag{6}$$

As shown in Figure 8, the new DenseNet network consists of dense block layers, efficient channel attention layers, averagepool, and a fully connected layer.

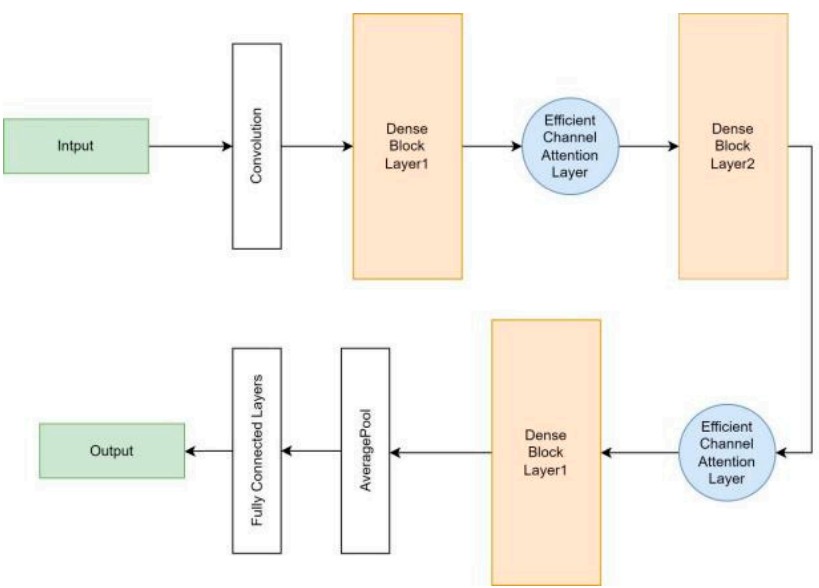

**Figure 8.** Schematic illustration of the overall architecture of the DenseNet (ECAD) network based on the attention mechanism.

*3.6. Lab Environment*

The experimental environment is presented in Table 1.

**Table 1.** Machine configuration information table.

| Projects | Content |
| --- | --- |
| Central Processing Unit | Intel (R) Core (TM) i7-7700K CPU @ 4.20 GHz (Santa Clara, CA, USA) |
| Memory | 32 G |
| Video card | NVIDIA GeForce GTX TITAN Xp (Santa Clara, CA, USA) |
| Operating System | Ubuntu 5.4.0-6ubuntu1~16.04.5 |
| CUDA | Cuda 8.0 with cudnn |
| Data Processing | Python 2.7, OpenCV, and TensorFlow |

*3.7. Evaluation Standard*

The following standard metrics were adopted here to measure the performance of the proposed system at different stages. It was defined as follows:

Dice Similarity Coefficient: This is a similarity measure between ground truth and predicted results.

$$DSC = \frac{2TP}{FP + 2TP + FN} \tag{7}$$

*Accuracy*: This is the proportion of the correctly predicted observations out of the total observations.

$$Accuracy = \frac{TP + TN}{TP + TN + FP + FN} \tag{8}$$

where

*TP* is defined as the positive sample predicted by the model as a positive class.
*TN* represents the negative samples predicted by the model as negative classes.
*FP* denotes the negative samples predicted by the model as positive.
*FN* refers to the positive sample predicted by the model as a negative class.

## 4. Results and Analysis

*4.1. Parameter Settings*

In experiments, a batch training approach was used to more effectively assess the discrepancy between the ground truth and the anticipated values. Other parameters include the following: loss function = cross-entropy loss; weight initialization method = Xavier; initialization bias = 0; initial learning rate of the model = 0.001; batch size = 16; momentum = 0.9; stochastic gradient descent (SGD) optimizer; and Softmax classifier. The model was reduced by 0.1 every 10 iterations. A total of 51 training epochs were completed, the input picture size was changed to 224 × 224, and the fused model was then combined with the final stored model.

*4.2. Segmentation Process Method*

In this part, the effectiveness of our suggested framework for automated segmentation and classification was assessed, and the findings are contrasted with the effectiveness of current approaches. The segmentation unit operates in two stages: only the multi-scale detection encoder–decoder network's performance was assessed in the first phase. The effectiveness was assessed in comparison to the techniques already in use, as indicated in Table 2. The encoder–decoder network and CRF module were merged in the second step, after which the performance was once again assessed. Only then were the results compared to the performance of the encoder–decoder system. Figure 9 compares the outcomes of the performance measures. The result was subsequently sent to a classification network for further processing.

**Table 2.** Segmentation performance comparison between our model and existing models.

| Techniques | Accuracy (%) | Dice Score (%) |
|---|---|---|
| Proposed model | 95.50 | 92.10 |
| FrCN [25] | 94.03 | 87.08 |
| CNN-HRFB [26] | 93.80 | 86.20 |
| FCN [26] | 92.70 | 82.30 |
| iFCN [27] | 95.30 | 88.64 |
| DCL-PSI [28] | 94.08 | 85.66 |

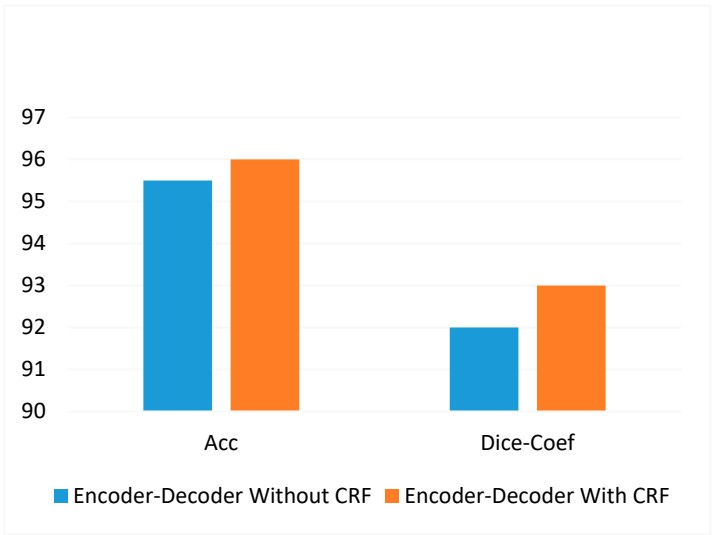

**Figure 9.** The figure compares the general performance of the encoder-decoder network using accuracy and Dice-coefficient when used with CRF and when used without CRF.

### 4.3. The Comparison of Different Classification Models

In order to verify the effectiveness of the model, different recognition models under the same experimental conditions were selected for comparative experiments. Since the standard deep convolutional neural network models VGG-16 [29], VGG-19 [29], ResNet50 [30], and DenseNet have achieved good recognition results in different fields, they were selected as the comparison model for this experiment. Table 3 shows the comparison results of each model test. The test accuracy and loss curves of the proposed model and the compared models are depicted in Figure 10.

**Table 3.** Classification experiment results of each model.

| Techniques | Size (MB) | Loss (%) | Test Accuracy (%) | Test Time/Image (ms) |
|---|---|---|---|---|
| VGG-16 | 800.33 | 35.45 | 81.98 | 153.30 |
| VGG-19 | 832.45 | 33.34 | 87.94 | 163.10 |
| ResNet50 | 95.23 | 19.33 | 90.74 | 104.50 |
| DenseNet | 75.43 | 29.34 | 91.37 | 98.80 |
| FCN-ECAD | 143.50 | 6.55 | 98.28 | 69.20 |

As can be seen from Table 3, the test accuracy of the VGG16 model was 81.98%, which was obviously not suitable for this experiment. The recognition accuracy of the VGG19 model was higher than that of the VGG16 model, mainly because of the three-layer convolution added by VGG19 on the basis of VGG16. Hence, it can be inferred that the performance was improved. However, the amount of calculation and model memory also increased. Compared with VGG, the ResNet50 model based on different sparse structure designs reduced a large number of model parameters, and the performance was also significantly improved. The test accuracy rate reached 90.74%. The improved FCN-ECAD

model in this work has a test accuracy rate of 98.28%, which was obviously better than other models, the model size was also small, only 143.5 MB, and the memory usage was very small.

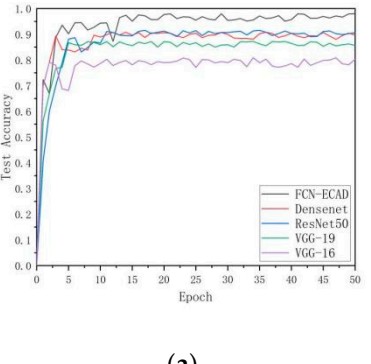

(**a**)

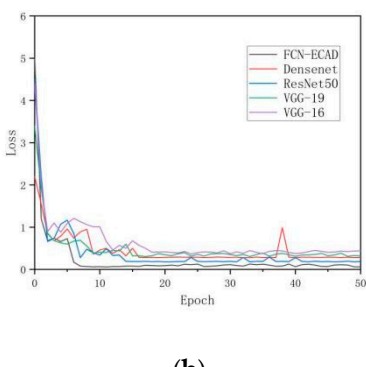

(**b**)

**Figure 10.** Recognition accuracy and loss curves of the comparison models: (**a**) model test accuracy comparison and (**b**) model loss comparison.

Table 3 also lists the detection time of the model in this work and other deep network models for the recognition of a single insect image. Each deep neural network model carried out 10 tests on a single image, and finally, the average test time was taken as the test detection recorded the results. As can be seen from the test results, VGG-19 required the longest time to detect a single image, with an average detection time of 163.1 ms. ResNet50 and DenseNet single-sheet detection times were 104.5 ms and 98.8 ms, respectively. The FCN-ECAD model proposed in this work only took an average of 69.2 ms to detect a single image, which was more suitable for the rapid detection of rice pest images.

According to the above-mentioned table, it can be concluded that under this model, the classification of rice pest data can obtain a high accuracy rate. As can be observed from Table 4, three insects, namely HolotrichiadiomphaliaBates, Gryllidae, and mole cricket, have a higher recognition success rate. This is because compared with other insects, the characteristics of these three insects are more obvious, and the characteristics can be easily extracted for recognition. The accuracies of armyworm and cutworm were relatively low. This is because the data sets of these two insects have similar appearance characteristics in some body shapes, and it was difficult to extract effective feature points for identification. The brown rice planthopper had the lowest accuracy rate, because compared to other insects, the brown rice planthopper was too small in size, the pixels in the original data were very low, and the feature points were difficult to extract. Therefore, the accuracy of the developed model was not high. In the future, the detailed extraction of feature points should be further improved.

**Table 4.** Classification results of different kinds of pests.

| Label | FCN-ECAD (%) | Vgg16 (%) | Vgg19 (%) | ResNet50 (%) | DenseNet (%) |
|---|---|---|---|---|---|
| *C. suppressalis* | 98.89 | 81.23 | 86.56 | 88.97 | 90.43 |
| *N. aenescens* | 98.99 | 80.38 | 87.88 | 89.99 | 91.58 |
| *C. medinalis* | 97.28 | 79.99 | 87.56 | 89.69 | 90.43 |
| *N. lugens* | 96.42 | 75.25 | 80.95 | 85.63 | 87.63 |
| *A. ypsilon* | 97.33 | 81.98 | 86.89 | 90.45 | 91.22 |
| *G.* sp | 99.53 | 85.65 | 92.33 | 94.21 | 94.38 |
| *M. separata* | 97.35 | 78.65 | 88.15 | 91.59 | 89.92 |
| *H. armigera* | 98.49 | 82.79 | 87.68 | 90.24 | 90.34 |
| Gryllidae | 98.98 | 86.78 | 91.58 | 92.99 | 93.87 |
| *H. diomphalia* | 99.58 | 87.11 | 89.87 | 93.65 | 93.99 |
| Average | 98.28 | 81.98 | 87.94 | 90.74 | 91.37 |

Insects of the same family tend to have similar morphological characteristics. Hence, it was difficult to classify them precisely. Compared with other models, as shown in Table 4, when the model in this work recognized insects with large differences, the model accuracy can reach about 99%. When identifying insects with inconspicuous characteristics and small differences, the accuracy of the model can also reach about 98%. The algorithm in this work had a high recognition accuracy rate in the recognition of similar insects, which demonstrates that the proposed algorithm has good robustness. It also has high performance for object classification with similar features.

### 4.4. Impact of ECA on Model Performance

The improvement of the model can be divided into two cases: ECA with an attention mechanism and ECA without an attention mechanism. The three aspects of recognition accuracy, model size, and detection time of a single image were also compared. The results are shown in Table 5.

**Table 5.** Model experiment accuracy with or without ECA.

| Techniques | Test Accuracy (%) | Size (MB) | Test Time/Image (ms) |
|---|---|---|---|
| Proposed model without ECA | 94.34 | 142.67 | 68.40 |
| Proposed model with ECA | 98.28 | 143.50 | 69.20 |

The improved model had a higher accuracy rate on the training set and test set, and whether there is an attention mechanism ECA in the model has a greater impact on the accuracy of the model. After the attention mechanism ECA was added, the model's accuracy rate increased from 94.34% to 98.28%, which was a significant improvement. In addition, the model size only increased by less than 8 MB. This is because, during the process of feature extraction, the incorporation of the ECA attention mechanism can effectively strengthen the extraction of complex background insect features, prevent the occurrence of overfitting phenomenon, ensure that the network learns effective feature information, and greatly improve the accuracy rate.

### 5. Conclusions

This research offers some new approaches in segmentation and classification methods for rice insect pest images by using deep learning techniques. Firstly, we introduced a new encoder–decoder in the FCN and a series of sub-networks connected by jump paths that combine long jumps and shortcut connections for accurate and fine-grained insect boundary detection. Secondly, the network also integrates a CRF module for insect contour refinement and boundary localization, and finally, a novel DenseNet framework that introduces an ECA is proposed, focusing on extracting insect edge features for effective rice pest classification. The proposed model was tested on the data set collected in this paper with a final accuracy of 98.28%, showing a better performance than existing methods. Moreover, the model in this paper also maintains high model accuracy with good robustness in the classification of small target insects and insects with the same physical characteristics, while it can be demonstrated from our results that effective segmentation of insect images prior to classification can improve the detection performance of deep-learning-based classification systems. This paper solves the problem of the rice insect pest classification and provides a theoretical basis for the subsequent research of agricultural pest identification.

**Author Contributions:** Conceptualization, H.G. and T.L. (Tonghe Liu); methodology, H.G.; software, T.L. (Tonghe Liu); validation, T.L. (Tianye Luo), J.G., and R.F.; formal analysis, J.L.; investigation, X.M.; resources, Y.M.; data curation, Y.S.; writing—original draft preparation, H.G.; writing—review and editing, S.L. and Q.W.; supervision, T.H.; project administration, Y.G. All authors have read and agreed to the published version of the manuscript.

**Funding:** This research was funded by the Changchun Science and Technology Bureau, funding number 21ZG27 (http://kjj.changchun.gov.cn) accessed on 1 July 2021, the Jilin Provincial Development and Reform Commission, funding number 2019C021 (http:/jidrc.jl.gov.cn) accessed on 1 January 2019, and the Department of Science and Technology of Jilin Province, funding number 20210302009NC (http://kjt.jl.gov.cn/) accessed on 1 November 2021.

**Data Availability Statement:** All new research data were presented in this contribution.

**Conflicts of Interest:** The authors declare no conflict of interest.

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
