# Peer review of "Based on FCN and DenseNet Framework for the Research of Rice Pest Identification Methods"

_agronomy, doi:10.3390/agronomy13020410_

Round 1

Reviewer 1 Report

This manuscript proposes an FCN-ECAD framework to detect and classify rice pest insects, and the proposed model outperforms other models. The segmentation and classification accuracies may be good; however, the manuscript was so poorly written in structure and grammar that I think it can not be published until it is thoroughly revised.

Some specific problems in the MS are shown below:

1.       The abstract, the introduction, and the conclusion should be rewritten logically, concisely, and clearly.

2.       The pest images in figure 1 are too casually listed. Please select some typical pictures to enlarge.

3.       How were the labels made? How to validate the accuracy of the segmentation results? Please state clearly.

4.       Sections 4.1 and 4.2 are not results and should not be placed in the results section.

5.       Captions of table 2 and table 3 should be more precise. For example, are they segmentation accuracy or classification accuracy?

6.       There are too many grammatical errors and poor handwriting in the manuscript, such as in Lines 32, 61, 125, 126, 223, 288, and so on. The first letter of the title needs to be capitalized. Unfortunately, there are too many such mistakes to list them all.

7.       The parameters in some equations were so nonstandard, such as the “Yi” in Line 178, which should be “Yi”, and the “Xl-1” in line 231, which should be “xl-1”. All these nonstandard writing makes me feel that the author is extremely careless in the process of writing.

8.       It would be best if you listed the full name when an abbreviation first appears.

9.       The same terms need to be consistent in the full text. Take the “DenseNet” for example, there are so many “DenseNet” and “Densenet”.

10.    The paragraph from lines 280 to 282 should be removed, which is the original words in the template.

Reviewer 2 Report

The authors provided a novel deep learning approach for segmentation and classification of rice pest images. The results are relevant to leverage the knowledge around the topic. However, the text language and description of information needs to have a massive improvement.

The introduction section is only talking about different accuracy results of methodologies in different agriculture systems and is not clear how this work makes the research on rice pest identification particularly important. What is the relevance of having an automated system for rice pest identification? This information is not included.

In addition, there almost no discussion of the results. The authors mainly described the results without contextualization.

Finally, the authors worked with 10 different insect species and some of them presents similar morphological characteristics. I highly recommend to have a taxonomist/entomologist report to confirm the identification at an insect level to support the classification outputs.

Below there are some suggestions for the text improvements.  

47 - Traditional machine vision to pest identification? What are the traditional methods?

51 - Convolutional neural network technology is not also deep learning?

55 - “as early as possible” is not a clear definition. Decision making for insect pest control needs to be in the right time and with appropriate environmental conditions.  

61 - “Yang et al.” Reference incomplete.

70 – “Compared with the traditional network, Sun et al.” Reference incomplete.

74 “Before the advent of FCN, there were also some traditional methods for semantic segmentation” What is FCN? Is the first time that you are mentioning this the text. Also, what is the context to using semantic segmentation? Why semantic segmentation is important?

81 “The second is that the network is not deep enough to extract more abstract semantic information” In CNN we can configure the deep of the network, isn’t it? So why this is a limitation?

83 “traditional CNN cannot accurately avoid irrelevant factors” What irrelevant factors? Please be more specific to make clear what you are suggesting for the reader.

92-95 – Why there are some results being explained here in the paragraphs? In addition, the importance and relevance of the study needs to be improved. For example, the authors concluded that “at the same time speeds up the efficiency of field insect automatic recognition, which has broad application prospects”, what is the time spent by manual identification? There was no comparison that can support this conclusion and what broad application prospects?

100-102- The sentence is not clear “insects are heated and inactivated by the electric heating plate to make it easier to shoot’? Please, be clear for the reader, perhaps what you are trying to say is: insects were transferred from the light traps to laboratory conditions to be prepared for the data collection? Then you describe the preparation methodology...

105 – “The industrial camera…”, what Camera? From what manufacturer and model?

107 – “The resolution is 4024 x 3036” 4024 x 3036? This is in pixel? What unit?

114 – “this experiment selects 10 kinds of rice that are common and large in number” select rice insect pests or difference rice genotypes? Please, be specific.

116-118- You need to standardize the description of the species; I suggest the following: Chilo suppressalis (Lepidoptera: Crambidae) then adopt the same description of specie and order to be consistent.

118-119 – “Since the trapping device is based on the phototaxis of insects, this article The identification of the study was directed to the adults” there are lots of typos in this sentence that needs to be fixed. Also, there are words in capital letters in the middle of the text that makes reading more difficult. Please, review this across the document.

123-126 – I think you need to review the description of the sample size in the specie level, how you can have 2,56 pictures? It should be 2 or 3 not decimal number?

151-155 – Again, it would be great to standardize the way you describe the numbers it’s confusing.

228 – Typo in the section description

Round 2

Reviewer 1 Report

The revised version is much improved in basic writing with far fewer low-level errors. The English language also has some improvement, but it still needs to polish further. As for the contents of this article, there still are some issues need to be addressed. Unfortunately, I don’t think this manuscript is suitable for publication in its present format. A major revision is still needed.

In the first round, I advised the authors to rewrite the abstract, the introduction, and the conclusion. I didn’t just mean the language. The organizational logic of these three parts also needs to be improved. Specifically: 

FCN is the basic framework of the prosed algorithm and it is an important part of the title, while it isn’t appeared in the abstract. The proposed method was called various names, such as algorithm, system, model, which makes readers confusing easily. Additionally, the descriptions of the proposed method should be more concise.

In the introduction, 1) the first paragraph and lines 61 to 67 should be merged, cause these two parts are all stating the practically demand. And I don’t think the identification of pests of this work is related to pest forecasting. 2) In lines 80 to 84, you stated some problems of the above mentioned models and said they were not suitable for rice pests. It seems like illogical. Are rice pests more difficult to identify than the above pests? If so, may be the different between them should be explained. 3) in the second paragraph from the bottom, the defects of the traditional methods for semantic segmentation are needless. You should state what FCN and DenseNet are, what’s their advantage, and why you must introduce the proposed method on the basis of FCN, and so on.

Some other specific problems or mistakes are list below:

 1. Line 21, it seems like it is improper to use “insufficient” to to with “size”.

2. Line 69-71, bad statement and error writing for Yang et al [6].

3. Line 113 and 115, unify the equipment name.

4. Line 127 and the caption of figure1, image should be plural.

5. All elements of all figures should be unified, including font style, font size, and font color. Font size should be not too big or too small.

6. Line 255, DenseNet, other than Densenet. Please be careful.

7. The explanation of parameters of equations in test should be italic, please check carefully.

8. In Line 278, 280, and 281, the parameters are not unified, still, even though I had point out this problem in the first round!

9. How did you make the ground truth to truth to validate the segmentation accuracy?

10. The title of section 4.3 should be “The comparison of different classification models”

11. Line 359-366 is not result and analysis, and the title is not precise.

12. FCN-ECAD first appears in Table 3, you should define it in the text.

13. Conclusions are too long-winded; they should be rewritten concisely.

Reviewer 2 Report

The manuscript showed considerable improvement after the considerations incorporated into the text. However, some aspects still need to be improved before being considered for publication.

1- The description of species needs to be improved throughout the text as follows: The overall recommendation is to describe the genus and species description only the first time that is being mentioned in the text. Then, you can abbreviate the description since you had detailed in the first time. Below you have an example. 
The first time that this specie is being mentioned in the text: Nilaparvata lugens. 
Then, the next time you can only describe by using: N. lugens.

Remember, the sentence need to be in italic format.

In addition, please, be consistent sometimes you describe the author who described the species and sometimes not, just choose one and standardize across the document (e.g. Naranga aenescens Moore (Lepidoptera:Noctuidae) change to Naranga aenescens (Lepidoptera:Noctuidae). Also, the authors need to review the descriptions to avoid miss writing (e.g. Helicoverpaarmigera change to Helicoverpa armigera; Gryllotalpaspps change to Gryllotalpa sp.)
